# Actin assembly ruptures the nuclear envelope by prying the lamina away from nuclear pores and nuclear membranes in starfish oocytes

Natalia Wesolowska[1], Ivan Avilov[2], Pedro Machado[3†], Celina Geiss[1], Hiroshi Kondo[1‡], Masashi Mori[1§], Peter Lenart[1,2]*

[1]Cell Biology and Biophysics Unit, European Molecular Biology Laboratory (EMBL), Heidelberg, Germany; [2]Max Planck Institute for Biophysical Chemistry, Göttingen, Germany; [3]Electron Microscopy Core Facility, European Molecular Biology Laboratory (EMBL), Heidelberg, Germany

*For correspondence:
plenart@mpibpc.mpg.de

Present address: †Centre for Ultrastructural Imaging, King's College London, London, United Kingdom; ‡The Francis Crick Institute, London, United Kingdom; §RIKEN Center for Developmental Biology, Kobe, Japan

Competing interests: The authors declare that no competing interests exist.

**Abstract** The nucleus of oocytes (germinal vesicle) is unusually large and its nuclear envelope (NE) is densely packed with nuclear pore complexes (NPCs) that are stockpiled for embryonic development. We showed that breakdown of this specialized NE is mediated by an Arp2/3-nucleated F-actin 'shell' in starfish oocytes, in contrast to microtubule-driven tearing in mammalian fibroblasts. Here, we address the mechanism of F-actin-driven NE rupture by correlated live-cell, super-resolution and electron microscopy. We show that actin is nucleated within the lamina, sprouting filopodia-like spikes towards the nuclear membranes. These F-actin spikes protrude pore-free nuclear membranes, whereas the adjoining stretches of membrane accumulate NPCs that are associated with the still-intact lamina. Packed NPCs sort into a distinct membrane network, while breaks appear in ER-like, pore-free regions. We reveal a new function for actin-mediated membrane shaping in nuclear rupture that is likely to have implications in other contexts, such as nuclear rupture observed in cancer cells.

## Introduction

The nuclear envelope (NE), composed of inner and outer nuclear membranes, is a specialized sub-compartment of the endoplasmic reticulum (ER) that separates the nucleus and the cytoplasm in eukaryotic cells. The inner and outer NE is fused at nuclear pore complexes (NPCs) to mediate nucleo-cytoplasmic transport. This complex NE membrane structure is mechanically supported by a network of intermediate filaments, the lamina, which lines the nucleoplasmic side (*Burke and Ellenberg, 2002*).

Across species and cell types a considerable diversity of nuclear structure allows adaptation to physiological function. For example, the composition of the lamina is adapted to provide the high mechanical stability that is necessary in muscle cells, or sufficient flexibility in immune cells, which need to squeeze through confined spaces (*Thiam et al., 2016*). Oocytes have a very specialized nuclear architecture with an exceptionally large nucleus, also known as the germinal vesicle, which stores nuclear components that are necessary to support early embryonic development. The oocyte NE is densely packed with NPCs that serve as a stockpile of these complexes (rendering oocytes a popular model in which to study NPCs), and the lamina is thick so that it is able to provide mechanical support for this very large structure (*Goldberg and Allen, 1995*).

The NE must be dismantled at the onset of every cell division to give microtubules access to chromosomes, and then reassembled at the end of division once the chromosomes are segregated.

Depending on the species and nuclear architecture, there is a broad diversity in disassembly mechanisms. In *Drosophila* and *Caenorhabditis elegans* embryos, the NE and the lamina remains partially intact during cell division, whereas in vertebrates and deuterostomes (including the echinoderm starfish), the complex NE structure is fully disassembled during division. In somatic mammalian cells, NE disassembly involves the complete dismantling of the NPCs, depolymerization of the lamina, and re-absorption of the nuclear membranes into the ER (*Hetzer, 2010*; *Ungricht and Kutay, 2017*).

In all species in which nuclear envelope breakdown (NEBD) has been investigated in detail, including somatic cells and oocytes from various species, NEBD begins with a partial permeabilization of the NE resulting from phosphorylation-driven disassembly of the NPCs and other NE components (*Dultz et al., 2008*; *Mühlhäusser and Kutay, 2007*; *Terasaki et al., 2001*; *Lénárt et al., 2003*; *Martino et al., 2017*; *Linder et al., 2017*). This allows proteins, and smaller dextrans up to ~70 kDa, to leak in or out of the nucleus (*Lénárt et al., 2003*). Furthermore, it is likely that the mechanical properties of the NE are affected, that is the NE is weakened and destabilized as a result of the phosphorylation of lamins and lamina-associated proteins (*Ungricht and Kutay, 2017*). Importantly, however, during this first phase of NEBD, the overall structure of the NE (as observed by electron microscopy (EM)) is still intact and the compartmentalization of large protein complexes (e.g. ribosomes and microtubules) is maintained (*Terasaki et al., 2001*; *Lénárt et al., 2003*).

In all of the species and cell types investigated to date, the slow, phosphorylation-driven weakening of the NE is followed by a sudden rupture of the NE leading to rapid and complete mixing of cyto- and nucleoplasm. As this dramatic change is easily visible, even by transmitted light microscopy, this second step is commonly identified as 'NEBD,' marking the transition between the prophase and the prometaphase of cell division. Observations from several cell types suggest that this sudden rupture requires mechanical force generated by the cytoskeleton. In cultured mammalian cells, microtubules tear the NE in a dynein-dependent process (*Beaudouin et al., 2002*; *Salina et al., 2002*). By contrast, we have shown recently that in the large oocyte nucleus, the actin cytoskeleton rather than the microtubule cytoskeleton is required for NE rupture. A transient F-actin 'shell' is polymerized by the Arp2/3 complex on the inner surface of the NE and membranes undergo complete rupture within two minutes of its formation (*Mori et al., 2014*). The rapid and dramatic reorganization of the NE during NEBD that is mediated by cytoskeletal forces, involving either microtubule-driven rupture in somatic cells or the F-actin shell in oocytes, has not been well understood.

Here, we use a combination of live-cell and super-resolution light microscopy, together with correlated electron microscopy, to capture these sudden changes in NE organization. We find that the F-actin shell is nucleated within the still-intact lamina and projects filopodia-like spikes into the nuclear membranes. The resulting nuclear membrane protrusions are free of NPCs, but are juxtaposed by NPC-dense clusters. Subsequently, these NPC-dense conglomerates invaginate and sort into the NPC-rich membrane network, while breaks appear on the pore-free regions.

## Results

### F-actin assembly causes reorganization of the nuclear membranes leading to rupture

We have shown previously that NE rupture, characterized by a wave-like entry of large cytoplasmic molecules into the nucleus, is mediated by a transient F-actin shell on the inner side of the NE, which is nucleated by the Arp2/3 complex (*Mori et al., 2014*). The F-actin shell first appears as an equatorial band of foci when the NE is still intact and impermeable to large dextrans (*Figure 1A*, 0 s). Approximately 30–45 s later, as the F-actin foci grow and intensify, merging to form a continuous F-actin shell, the first breaks on the NE appear, allowing a large 500 kDa dextran to flood into the nucleus (*Figure 1A*, 45 s). The shell then spreads towards the poles before a wave of membrane rupture takes place after a delay of ~30 s (*Figure 1A*, 90 s).

It became obvious to us that we needed to address what happens in the 30–45 s period between the start of actin assembly and NE rupture, as this appears to be a critical moment in F-actin-driven NE rupture. Our previous live-cell imaging assays lacked sufficient resolution to tackle the question (*Mori et al., 2014*), therefore we established brighter, recombinant-protein-based probes and used

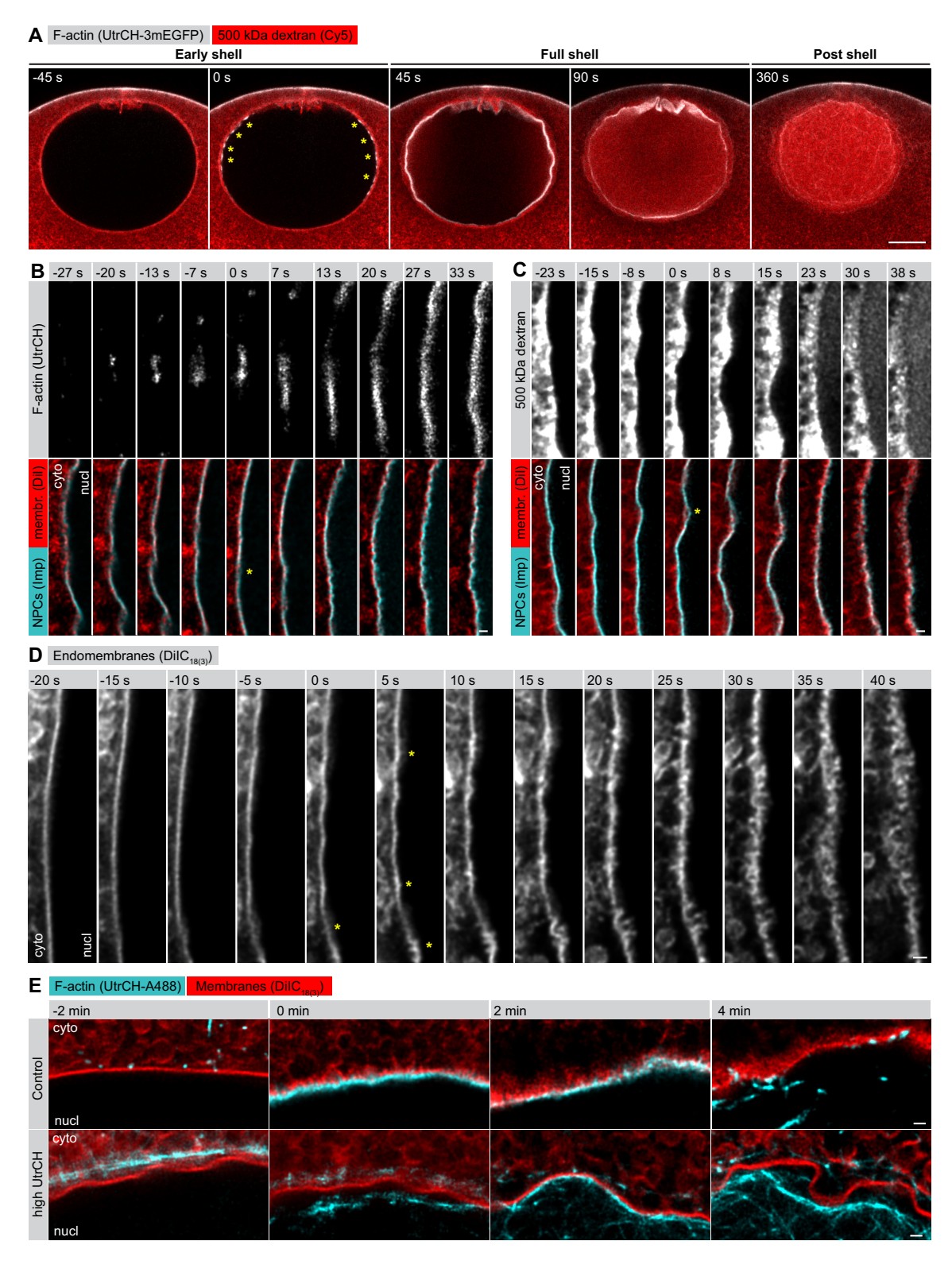

**Figure 1.** F-actin-driven membrane reorganization precedes NE rupture. (**A**) Live imaging of UtrCH-3mEGFP (white) and Cy5-labeled Dextran-500 kDa (red) in a starfish oocyte undergoing NEBD. Selected single confocal sections are shown from a time series. scale bar: 20 μm. Yellow asterisks mark F-actin foci. (**B**) Live imaging of a section of the rupturing NE in an oocyte injected with recombinant AlexaFluor488-labeled UtrCH, AlexaFluor647-labeled importin-β (45-462) (Imp), and DiIC$_{18(3)}$ (DiI). Selected frames are shown from a deconvolved AiryScan time series. Scale bar: 1 μm. Yellow

*Figure 1 continued on next page*

*Figure 1 continued*

asterisks mark the first visible sites of membrane reorganization. (**C**) As in panel (**B**) except that the oocyte was injected with a 500 kDa dextran instead of UtrCH. (**D**) Similar to panel (**B**) except that the oocyte was injected with DiIC$_{18(3)}$ alone and imaged at the highest possible frame rate and resolution. (**E**) In an experiment similar to that shown in panel (**B**), oocytes were injected with either low (~1 µM) or a very high amount (~20 µM) of UtrCH-AlexaFluor488 and equal amounts of DiIC$_{18(3)}$. At high concentrations, UtrCH depletes the available actin monomers by stabilizing cytoplasmic F-actin networks, and thereby efficiently prevents Arp2/3-driven assembly and NE rupture.

the AiryScan technology combined with deconvolution to visualize changes in NE reorganization at the sub-micrometer scale.

When we looked at the process with our new approach, we saw that as the first F-actin foci began to assemble, the NE still appeared smooth and continuous, as visualized by a membrane probe (DiIC$_{18(3)}$) and a marker of NPCs (a fragment of importin-β) (*Figure 1B*, −13 s). 10–15 s later, we observed the first spike-like membrane protrusions towards the cytoplasm, which then rapidly escalated into a complex, convoluted and fragmented membrane structure (*Figure 1B–D*). Intriguingly, during this process NPCs appeared to separate partially from the membrane towards the nucleoplasmic side. Dextran entry did not coincide with the formation of the initial membrane spikes, it occurred 20–30 s later, when membrane reorganization had progressed to a rather advanced stage (*Figure 1C*).

This membrane reorganization is dependent on actin assembly. If the F-actin shell is inhibited by injecting a large amount of the actin-binding UtrCH domain to deplete available actin monomers, the NE membrane stays smooth and continuous (*Figure 1E*). Even in this case, the nucleus does collapse and the NE ruffles and folds as a result of the progressing partial NE disassembly. However, unlike in control oocytes, these large and folded membrane fragments remain intact long after NEBD, as we showed in earlier work (*Mori et al., 2014*).

Thus, we concluded that, in starfish oocytes, actin assembly mediates a reorganization of nuclear membranes that leads to NE rupture.

## Lamina remains intact during NE rupture

We next turned to immunofluorescence to visualize the endogenous NE components at an even higher resolution in order to reveal fine details of the F-actin-mediated NE rearrangements. We developed an antibody against the only identified starfish lamin protein, which together with the pan-NPC antibody mAb414 enabled us to visualize endogenous NE components together with phalloidin-stained F-actin. However, the F-actin shell is very transient, polymerizing and depolymerizing within 2 min, so the development of a reliable temporal reference for fixed-cell assays was also necessary. Fortunately, the F-actin shell emerges in a highly reproducible spatial pattern, which enabled us to time the fixed samples by correlating them with morphologies observed live (compare *Figures 1A* and *2A*).

With this assay in hand, we first wanted to clarify whether the F-actin shell destabilizes the NE by tearing the lamin network, as this mechanism has been reported for microtubule-driven NE rupture in somatic cells (*Beaudouin et al., 2002*; *Salina et al., 2002*). We have addressed this question in starfish oocytes before, but in this earlier work, we exogenously overexpressed a GFP fusion of human lamin B, which could have had different disassembly kinetics (*Lénárt et al., 2003*). To visualize the endogenous lamina directly this time, we analyzed samples stained with the starfish anti-lamin antibody at different stages of NE rupture. This confirmed our previous observations that the endogenous lamin network remains continuous even minutes after NE rupture, although it folds and ruffles as the nucleus collapses during NEBD (*Figure 2B*). In addition, imaging portions of the lamina *en face* by stimulated emission depletion (STED) microscopy suggests that the lamin mesh gradually coarsens during the process of NEBD (*Figure 2B*).

We conclude that the rupture of the NE does not proceed by F-actin-induced tearing or rapid disassembly of the lamina, which remains a continuous network throughout NEBD.

## The F-actin shell assembles within the lamina sprouting spikes that separate nuclear membranes

In order to localize the F-actin shell relative to NE components, we next co-localized the lamina or the nuclear membranes (as marked by NPCs) at the time of shell formation, with the F-actin shell

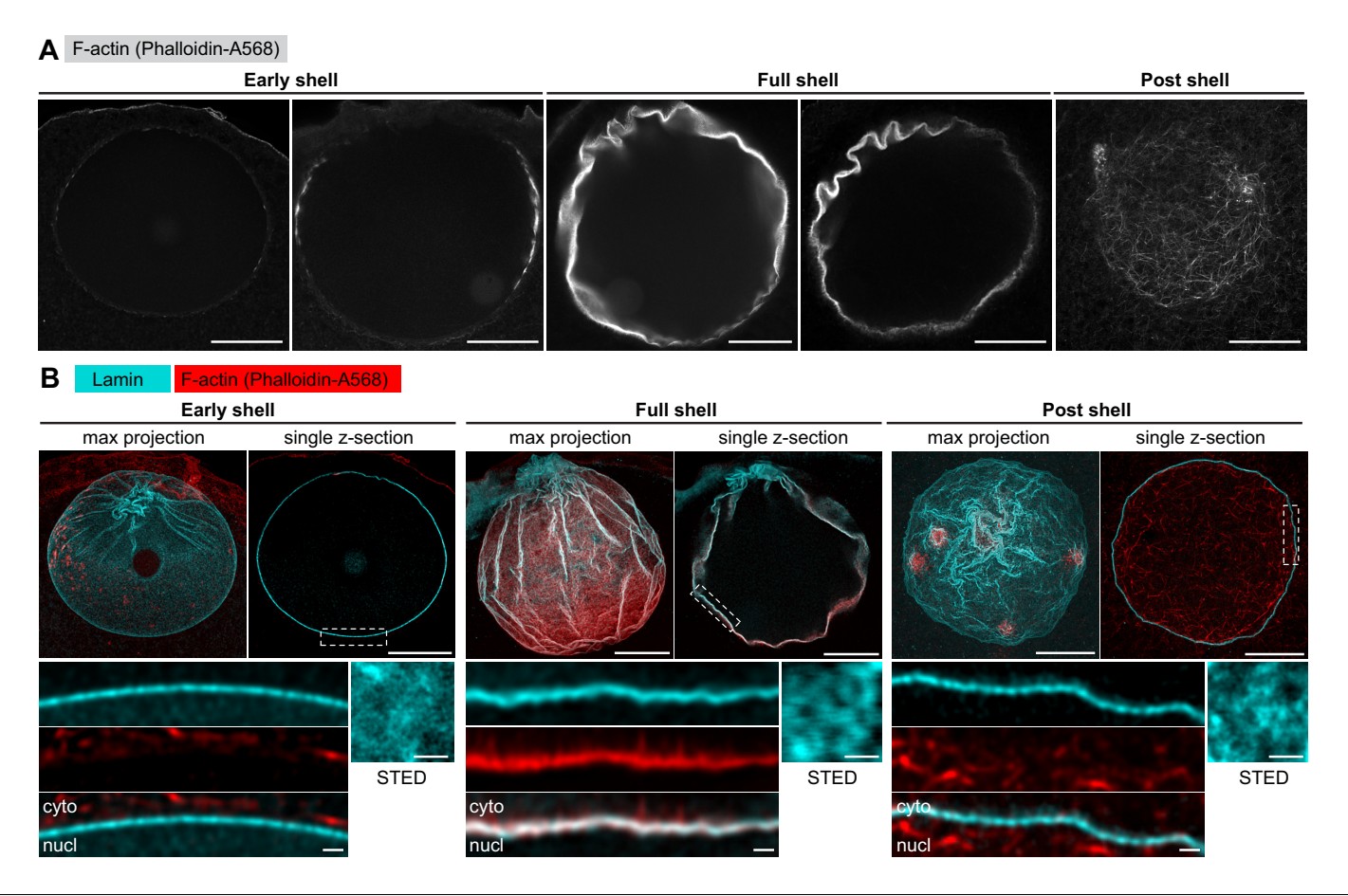

**Figure 2.** The lamina remains intact during NE rupture. (**A**) Fixed samples with F-actin labeled by phalloidin-AlexaFluor568. Individual confocal sections are shown and ordered to match the live time series shown in *Figure 1A*. Scale bar: 20 μm. (**B**) Immunostained starfish oocytes with anti-lamin antibody shown in cyan and phalloidin-AlexaFluor568 in red. The images show three time-points: early shell, full shell and post shell. Each panel shows a maximum projection of the whole z-stack (left), a single selected optical section across the equatorial plane (right), and a close-up of the area in the single section highlighted with dashed rectangle (bottom). The small insets on the bottom right show *en face* views of the lamina in oocytes stained with the anti-lamin antibody and imaged by stimulated emission depletion (STED) microscopy at the corresponding stages. Scale bars: 20 μm (top), 5 μm (bottom left) and 0.5 μm (bottom right).

stained by phalloidin. We observed that although the lamina co-localized with phalloidin, the NPC staining formed a separate layer of fragmented appearance up to 500 nm 'above' the F-actin shell (*Figure 3A,B*). Thus, the still-intact lamina appears to serve as the scaffold upon which the F-actin shell assembles, whereas the nuclear membranes appear to fragment and separate away from the lamina and the F-actin shell.

We confirmed these observations by STED imaging of the lamina and NPCs (note that these samples were co-stained with phalloidin for staging, but it was not possible to image the third color on this particular STED setup). Before NE rupture, the NPC and lamina stainings overlapped at the approx. 50 nm resolution afforded by STED, which is consistent with the known ultrastructure of the NE (*Figure 3C*, pre-NEBD) (*Burke and Ellenberg, 2002*). In stark contrast, at the shell stage, we were able to visualize NPC-stained NE fragments 'floating' above the intact lamina (*Figure 3C*, post-NEBD). This separation of lamina and membranes is dependent on the F-actin shell, because this process did not occur when we prevented F-actin shell formation by inhibiting Arp2/3 using the small molecule inhibitor CK-666 (*Figure 3C*, post-NEBD + CK-666). Consistent with the live imaging described above, the NE appears ruffled due to collapse of nuclear volume even if the F-actin shell is inhibited.

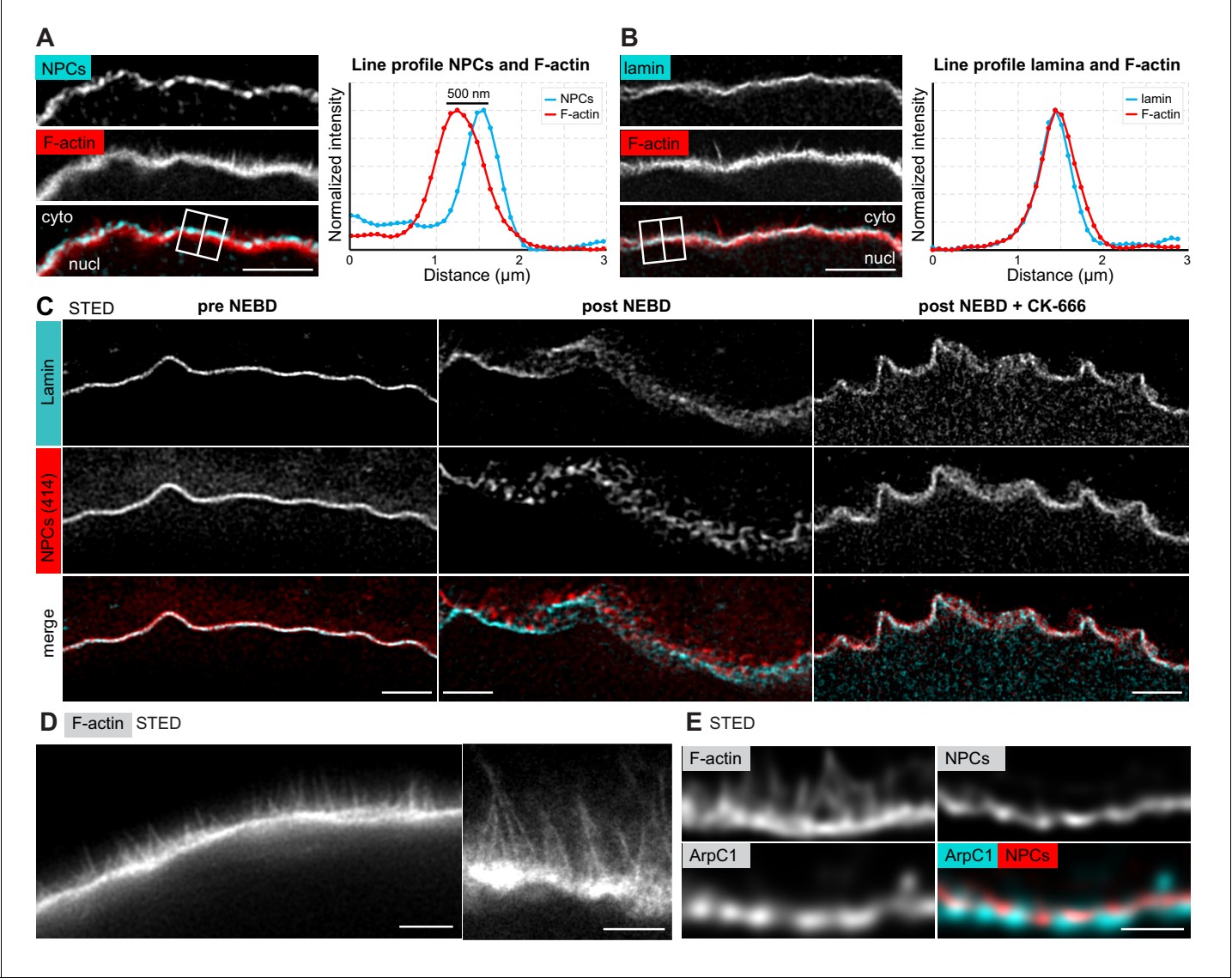

**Figure 3.** The F-actin shell separates the lamina and the nuclear membranes during NEBD. (**A**) Left: portion of the NE undergoing rupture immunostained with mAb414 for NPCs (cyan) and phalloidin-AlexaFluor568 for F-actin (red). A crop of a portion of NE from a confocal Z-section is shown. Scale bar: 2 µm. Right: plot of a line profile over the region marked with a white rectangle; normalized intensities of both channels are shown. (**B**) Same as panel (**A**) but stained with anti-lamin antibody and phalloidin-AlexaFluor568. (**C**) Portions of the NE stained with anti-lamin antibody (cyan) and mAb414 (red) and imaged by STED. Left, before NEBD; middle, after NEBD; right, after NEBD but first treated with CK-666 to inhibit the formation of the F-actin shell. Scale bars: 2 µm. (**D**) Phalloidin-Abberior Star RED staining of the F-actin shell imaged by STED microscopy. Scale bars: 5 µm (left) and 2 µm (right). (**E**) F-actin shell stained with mAb414 to label NPCs, an anti-ArpC1 antibody to label the Arp2/3 complex, and Phalloidin-Abberior Star RED to label F-actin, imaged by STED microscopy. Scale bar: 2 µm.

STED imaging of samples that were optimally fixed for phalloidin staining revealed filopodia-like F-actin spikes of 0.5–2 µm in length, spaced at ~0.1 µm, and extending from the base of the F-actin shell towards the nuclear membranes (*Figure 3D*). Furthermore, using a starfish-specific anti-ArpC1 antibody, we localized the Arp2/3 complex at the base of the F-actin shell just beneath the NE, whereas the F-actin spikes extending from the base were not labeled by ArpC1 (*Figure 3E*).

Taken together, our data show that the F-actin shell is nucleated by the Arp2/3 complex localized in the lamina and extends filopodia-like spikes, which separate the nuclear membranes away from the lamina.

## Correlative EM captures intermediates of NE rupture

Unfortunately, although live imaging showed dramatic rearrangement of membranes, we were unable to visualize fine membrane structures in immunofluorescent samples directly, because preserving F-actin in fixed oocytes requires the addition of detergents to the fixative. The oocyte NE is, however, densely packed with NPCs, so NPC staining does provide a good proxy for the nuclear membrane, as shown above. Nonetheless, during NE rupture, this organization may change. Therefore, to clarify the F-actin mediated rearrangements of nuclear membranes, we decided to target the early stages of the F-actin shell formation using electron microscopy (EM). In this time-window of approximately 30 s, when the actin shell has just partially propagated, we expected to observe intermediate steps of NE rupture, with parts of the NE already ruptured while other regions are still intact (see *Figure 1A*).

For this purpose, we developed a correlative electron microscopy protocol using high-pressure freezing and freeze substitution, which resulted in excellent preservation of the cellular structures (*Burdyniuk et al., 2018*). Correlation to light microscopy was achieved by using fluorescently labeled dextrans, which are directly visible on EM sections, as indicators of NEBD progression (*Figure 4A,B*): the small, 25-kDa dextran enters the nucleus during the first phase of NEBD through the disassembling NPCs, whereas the large, 160-kDa dextran only enters when the NE is ruptured (*Lénárt et al., 2003*). Thus, the stage when the 25-kDa dextran almost completely fills the nucleus but the large, 160-kDa dextran is still excluded identifies the time-window of F-actin shell formation and NE rupture.

We then performed automated large field-of-view montage transmission EM imaging of the whole nuclear cross-section to assess the state of the NE. An overview is shown in *Figure 4C*, this and two additional montages are available at high resolution as Supplemental Data (*Figure 4—figure supplements 1–3*). The montage illustrates the key advantage of the system, which allows the progression of NE rupture to be observed spatially ordered on a single section of the large oocyte nucleus. The arrangement of the rupture site is fully consistent with the live and fixed light microscopy data: NE rupture initiates near the 'equator' of the nucleus and spreads as a wave towards the poles (*Figure 4D*).

We carefully examined these large montages and observed a set of frequently recurring characteristic membrane configurations. We assigned them to one of four categories and gave each a symbol to mark their incidence (*Figure 4C*). Numbers under each category quantify the occurrence of each feature within the section, and numbers in parentheses represent similar quantifications in two other sections shown in *Figure 4—figure supplements 2* and *3*, illustrating that these are structures that occur frequently at the time of NE rupture.

Taken together, using our correlative light and electron microscopy approach, we were able to capture oocytes in the process of NE rupture.

## F-actin spikes protrude pore-free nuclear membranes

The large dataset (*Figure 4—figure supplements 1–3*), the high frequency of events observed and, importantly, the spatial arrangement from the equatorial rupture site towards the still-intact poles allowed us to reconstruct the steps of NE rupture and to correlate these to observations made in live and fixed cells.

First, as consistent with earlier observations, the NE is smooth, continuous and is tightly packed with NPCs with a regular spacing of ~200 nm in immature oocytes, as well as in oocytes just before NE rupture and even in the intact polar regions of the NE undergoing rupture (see *Figure 5A,B* for image examples, and *Figure 6C* for quantification) (*Lénárt et al., 2003*). By contrast, in areas closer to the rupture site, we observed regions with gaps in NPC occupancy, the number and size of which increased proximal to this region (*Figure 5C*). In the vicinity of the site, gaps appeared to evolve into 'bumps' and membrane spikes (*Figure 5D*). Although reconstructing the full 3D architecture of these spikes is challenging, even in 300-nm-thick tomographic sections, the most prominent spikes that we observed rose ~1 μm above the level of the NE (*Figure 5D,E*). These spikes contain fibrous densities consistent with actin filaments (*Figure 5D* and *Figure 5—figure supplement 1A*) and are even more clearly distinguishable on tomographic reconstructions (*Figure 5E*).

Intriguingly, on both thin sections and tomographic reconstructions, we observed that these spikes were covered by continuous nuclear membranes and almost completely free of NPCs

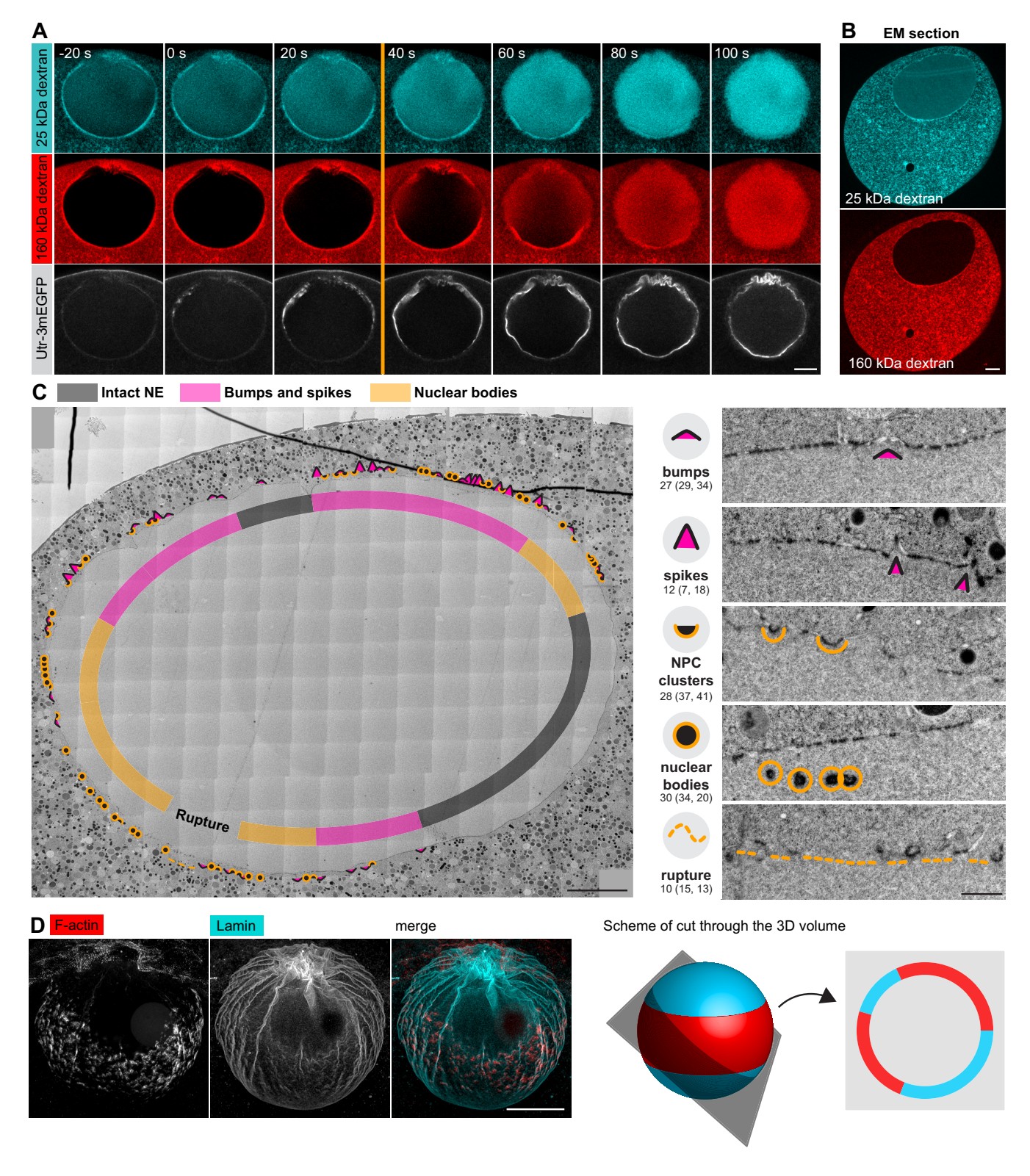

**Figure 4.** Correlative EM approach captures NE rupture intermediates. (**A**) Live imaging of a starfish oocyte undergoing NEBD and injected with a 25-kDa Cy5-labeled dextran (cyan), a 160-kDa TRITC-labeled dextran (red) and UtrCh-3mEGFP (white).Selected 2-frame projections from a confocal time-series imaged at one frame per second are shown. The orange line marks the moment immediately before rupture of the membrane, corresponding to the predicted time for the EM sample shown in panels (**B**) and (**C**). Scale bar: 20 μm. (**B**) Wide-field fluorescence image of a 70-nm section of a Lowicryl-

*Figure 4 continued on next page*

Figure 4 continued

embedded oocyte undergoing NEBD and injected with a 25-kDa Cy5-labeled dextran (cyan) and a 160-kDa TRITC-labeled (red) dextran. Scale bar: 20 µm. (C) A whole-nucleus tile of transmission EM images stitched automatically for the oocyte section shown in panel (B). Symbols around the nucleus correspond to NE rupture intermediates. A symbol legend with examples (crops from the tiled image) is shown to the right. Under each symbol, numbers correspond to the count of these events in the section shown, and the count in two adjacent sections is given in parentheses (shown in *Figure 4—figure supplements 1–3*). The band that traces the NE within the nuclear space demarcates areas with color-code for predominant membrane features. For a full resolution image, see *Figure 4—figure supplement 1*. Scale bar: 10 µm. (D) An oocyte fixed and stained with anti-lamin antibody (cyan) and phalloidin-AlexaFluor568 at early shell formation. A maximal Z projection is shown. Right: scheme illustrating the 3D geometry of the EM section. Scale bar: 20 µm.

The online version of this article includes the following figure supplement(s) for figure 4:

**Figure supplement 1.** High-resolution transmission electron microscropy (TEM) montage of a section through the nuclear region of an oocyte undergoing NEBD (shown on *Figure 3C*).

**Figure supplement 2.** High-resolution TEM montage of a section through the nuclear region of an oocyte undergoing NEBD, section adjacent to that shown in *Figure 4—figure supplement 1*..

**Figure supplement 3.** High-resolution TEM montage of a section through the nuclear region of an oocyte undergoing NEBD, section from the oocyte also shown in *Figure 4—figure supplement 1*. Scale bar: 10 µm.

(*Figure 5D*). These pore-free areas are surrounded by NPC-dense adjacent regions (*Figure 5D* and *Figure 5—figure supplement 1A,B*). We are unable to estimate the 3D membrane areas from 2D sections, nevertheless, our data suggest that NPC-free areas pushed out by the F-actin shell may cause the crowding of NPCs in juxtaposed regions (*Figure 6C*).

These observations are further supported by light microscopy. AiryScan imaging in live oocytes shows membrane-covered spike-like F-actin protrusions extending from the NE, and these protrusions are consistent in size and number with our observations by EM (*Figure 5F*). In fixed samples imaged by STED, we were unable to visualize membranes directly, but we observed F-actin spikes extending out of the lamina with no NPC staining covering them except for dense NPC clusters at their base, an arrangement that is fully consistent with that seen on EM samples (*Figure 5G*). In addition, we were able to visualize that the lamina runs at the base of the F-actin shell, below but still attached to NPC clusters, suggesting that NPCs might be held back at the base of spikes by their attachments to the lamina (*Figure 5G*).

Together, our data from EM correlated with live cell and immunofluorescence suggest that the growing gaps between NPCs, which then develop into bumps and spikes, are protrusions generated by Arp2/3-driven actin polymerization, which pries nuclear membranes and lamina apart. Membrane evaginations are largely free of NPCs, most probably because NPCs are still attached to the lamina, and thus NPCs are held back and cluster at the base of spikes.

## NPC-dense clusters invaginate and segregate from NPC-free regions

In regions closer to rupture sites, we observed an increasing segregation of pore-free and pore-dense segments (*Figure 6A*). Furthermore, accompanying spikes and NPC-rich clusters, we observed additional frequent membrane structures, which we call nucleoplasmic bodies. These are dense, round structures 200–500 nm in diameter beneath the NE in the nucleoplasm (*Figures 5D* and *6B*). These bodies often appear in a beads-on-a-string arrangement with a slightly electron-denser material connecting them (*Figure 6B*). Above them, the NE appears to be still intact, consisting mostly of NPC-free nuclear membranes and maintaining the nucleo-cytoplasmic boundary, as judged by the distribution of ribosomes. In pore-dense segments, as well as in nucleoplasmic bodies, our quantification shows that pore-to-pore distance decreased substantially from ~200 nm in intact areas to ~100 nm (*Figure 6C*), suggesting that the NPC-free membranes are generated by redistribution of NPCs.

Careful examination of intermediates suggests that nucleoplasmic bodies may form by invagination of NPC-rich clusters, initially inducing pits that curve into the nucleoplasm and then forming inverted NE tubules filled inside with cytoplasm (*Figure 6A,B* and *Figure 5—figure supplement 1B*). Sections capturing these structures *en face* and tomograms confirm that the electron density along the boundary of the bodies corresponds to closely juxtaposed NPCs with an intact central ring structure, similar to NPCs in still-intact areas (see 'nucleoplasmic bodies' in *Figure 6D,E*).

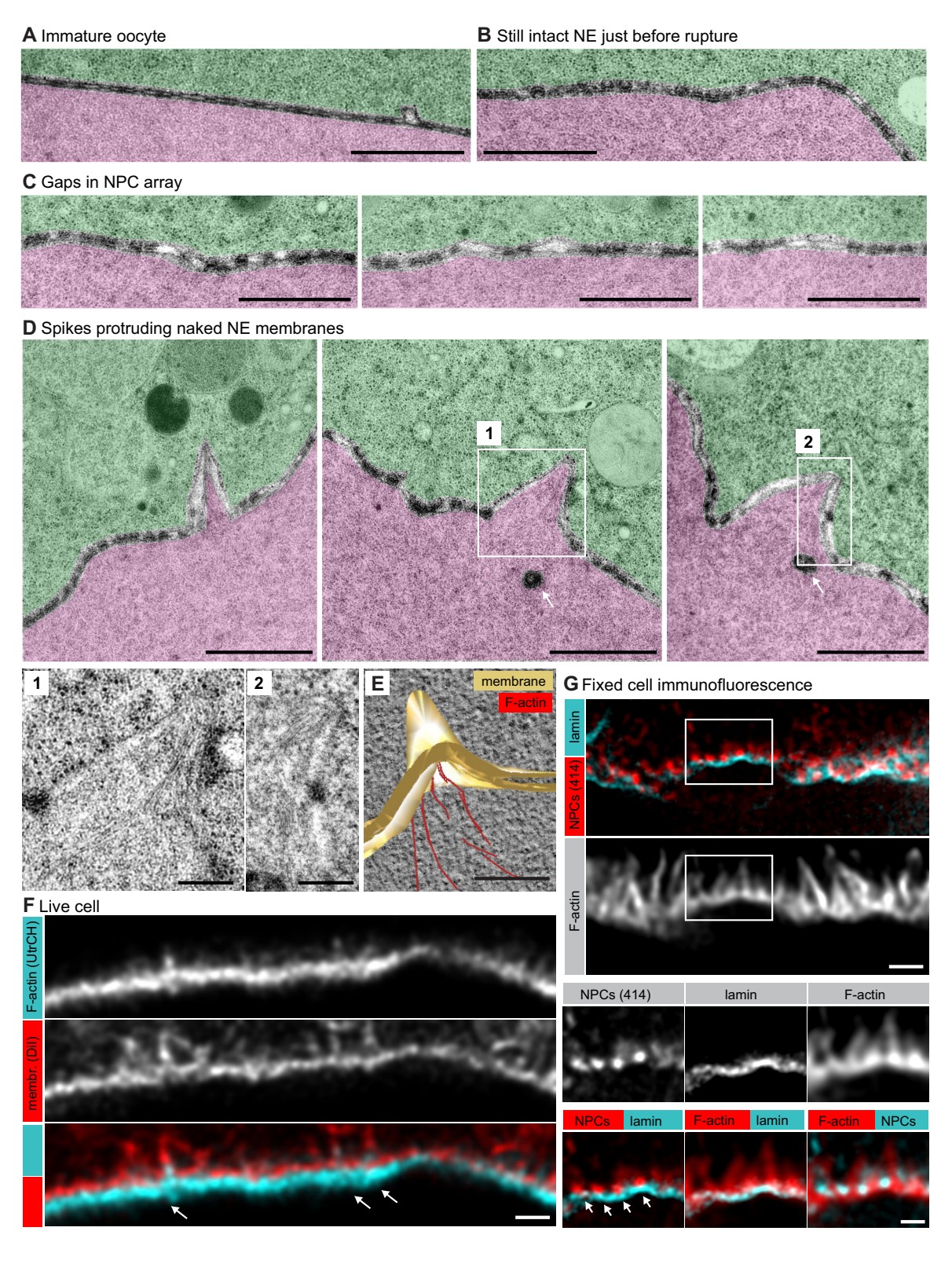

**Figure 5.** Spikes protrude bare nuclear membranes. (**A–D**) Transmission EM images from the oocyte shown in *Figure 4C* showing intermediates of NE rupture. Transparency coloring distinguishes the cytoplasm (green) from the nuclear area (pink) on the basis of the presence of ribosomes. Scale bars are 1 µm, except in the zooms, where they are 250 nm. In panel (D) arrows point to nucleoplasmic bodies. Zooms of the areas outlined with white rectangles are shown below. (**E**) Tomogram of a NE spike from a 300-nm thick section of the oocyte shown in *Figure 4C*, with model overlay

*Figure 5 continued on next page*

*Figure 5 continued*

segmented manually. Scale bar: 200 nm. (**F**) Selected single frame from a deconvolved 3D Airyscan recording at the time of NE rupture. F-actin is labeled with recombinant UtrCH-AlexaFluor488, and endomembranes are stained by DiIC$_{18(3)}$. Arrows mark prominent F-actin spikes. Scale bar: 1 µm. (**G**) STED image of the NE at the shell stage stained for NPCs (mAb414), lamina and phalloidin-Abberior Star Red. Separate channels and overlays are shown in the combinations indicated. Arrows point at nucleoplasmic bodies. Scale bars: 1 µm and 0.5 µm.

The online version of this article includes the following figure supplement(s) for figure 5:

**Figure supplement 1.** Examples of NPC-rich clusters.

Comparing these structures to membrane intermediates observed in live cells suggests that they correspond to the convoluted morphology seen just before rupture (*Figure 6F*). In fixed samples, large spots of NPC accumulations that appear to be attached to the underlying lamina are also strongly reminiscent of the nucleoplasmic bodies (*Figure 6G*).

Finally, the EM montages revealed that the membranes were interrupted in areas where the segregation of pore-dense and pore-free regions was strongest and NPC invaginations frequent and most dramatic (*Figure 7A*). Here, no continuous membrane boundary was seen to separate cyto- and nucleoplasm, but nucleoplasmic bodies were abundant. Light microscopy of the corresponding stage revealed a complex tubular-vesicular network that was densely labeled by NPC staining, occasionally resolving cross-sections of tubules consistent with our EM data (*Figure 7B*).

Together, our data show that NE rupture proceeds by F-actin-driven sorting of NE membranes into pore-dense and pore-free segments. Pore-dense segments invaginate, forming a network of nucleoplasmic bodies, whereas rupture occurs in evaginating pore-free segments.

## Discussion

Here, we resolved structural intermediates of rapid NE rearrangements mediated by the transient F-actin shell by using live-cell imaging, super-resolution light microscopy and correlative EM in starfish oocytes. On the basis of our data, we propose the following model for NE rupture (*Figure 7D*).

The first step is the formation of F-actin foci within the lamina. We hypothesize that these foci form at the time when cytoplasmic components, such as the Arp2/3 complex and actin monomers, reach a critical concentration in the nucleus as a result of the gradual, phosphorylation-driven disassembly and increasing leakiness of the NPCs in the first phase of NEBD. Once triggered, F-actin foci grow rapidly, which is expected as a result of the autocatalytic nature of Arp2/3-mediated nucleation of branched F-actin networks. As the foci spread, they merge to form a continuous shell. The filaments seem to grow preferentially from the shell base in the lamina towards nuclear membranes, and push against them. This asymmetry may be explained by the fact that force imposed on actin filaments promotes the nucleation of a branched meshwork (*Bieling et al., 2016*). Intriguingly, F-actin networks that are nucleated in vitro on micropatterned activated Arp2/3 show a morphology that is strikingly similar to that of the F-actin shell, with filopodia-like bundles pointing away from a base of dense branched network (*Reymann et al., 2010*). This suggests that localized activation of Arp2/3 within the lamina may be sufficient to explain the morphology of the F-actin shell.

Our light and electron microscopy data clearly show that the F-actin shell protrudes pore-free nuclear membranes, separating these from the lamina. We propose that these membranes are cleared of NPCs, because most NPCs are attached to the still-intact lamina at this stage, and thus are held back, while membranes are protruded by actin assembly. Then, as the NPCs accumulate between pore-free spikes, the membranes in these NPC-rich dimples buckle into the nucleoplasm and invaginate to form nucleoplasmic bodies. Our data suggest that pore-free nuclear membranes that are separated from the lamina are unstable, and thus rupture and rearrange into an ER-like reticular structure.

The exact mechanism of rupture remains unclear. However, in agreement with our model, it has been shown in various other physiological contexts that detaching the nuclear membranes from the lamin network leads to NE rupture. For example, NE rupture frequently occurs in cancer cells, in particular in micronuclei, where breaching of the NE barrier is preceded by local lamin disruption (*Hatch et al., 2013*). Recent work shows that the 'breakability' of the nucleus in migrating cells is dependent on lamin composition (*Thiam et al., 2016*; *Davidson and Lammerding, 2014*; *Denais et al., 2016*). Indeed, in somatic cells, it has been shown that NEBD proceeds by

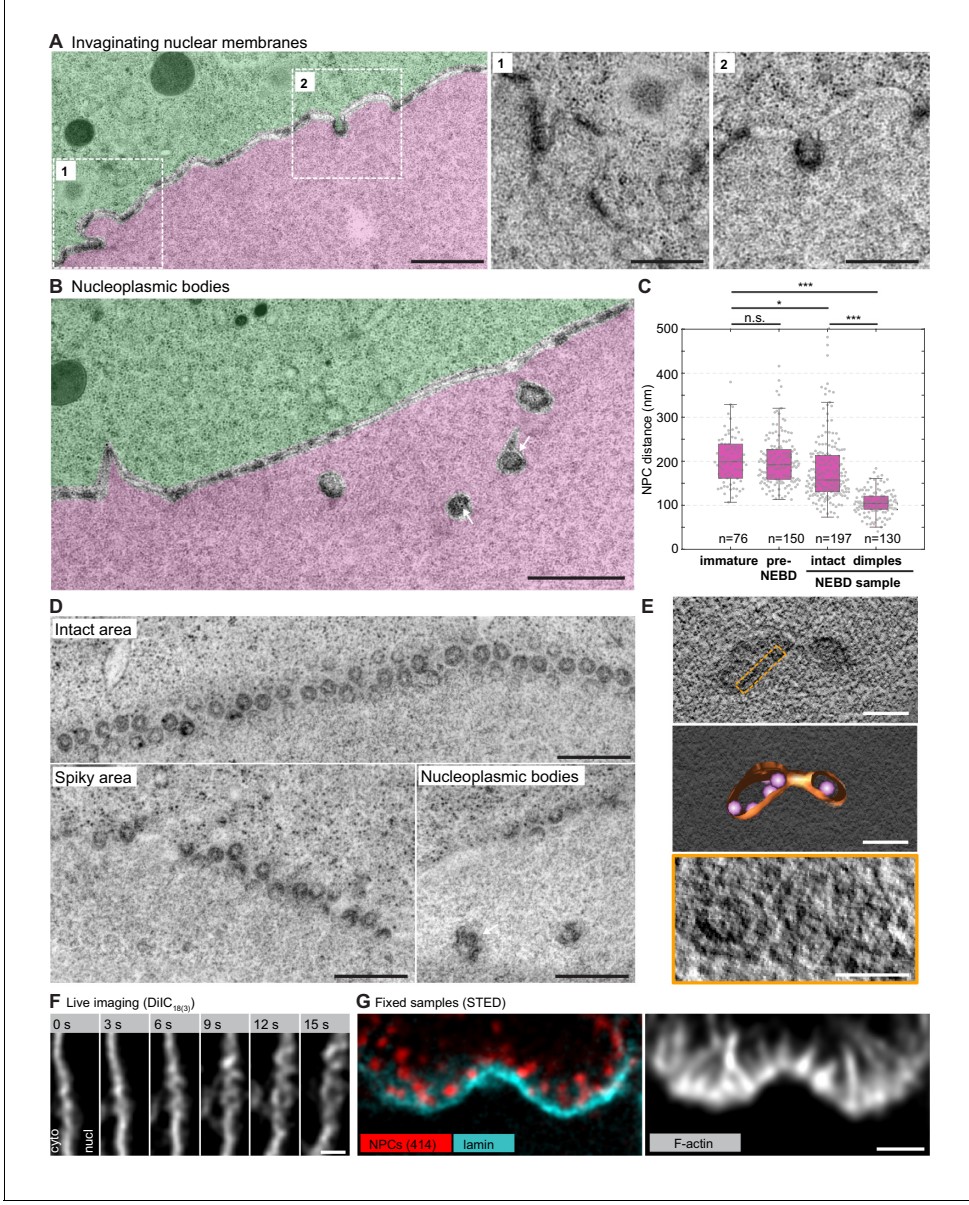

**Figure 6.** NPC clusters invaginate to form nucleoplasmic bodies. (A) Transmission EM images from the oocyte shown in *Figure 4C* and colored as in *Figure 5A–D*, showing invaginating NPC clusters. Zooms of portions outlined with white rectangles are shown below without color transparencies. Scale bars: 1 µm (left) and 500 nm(zooms). See *Figure 5—figure supplement 1A* for more examples. (B) Transmission EM images as in panel (A) showing an area with nucleoplasmic bodies. Arrows point at ribosomes that are present within nucleoplasmic bodies. Scale bar: 1 µm. (C) Quantification of pore-to-pore distance on electron micrographs similar to those shown in *Figure 4—figure supplements 1–3*. n refers to the number of pore-pairs measured in each of the respective samples or regions. n.s. denotes no significant difference, *, p≤0.05, ***, p≤0.001, as calculated by Student's T-test. (D) *En face* views of NPCs in the indicated regions along the NE. Scale bars: 500 nm. (E) Tomogram of a nucleoplasmic body (left), with a model overlaid (middle). Right: re-slicing of the volume perpendicular to the view on the left corresponding to the area outlined with an orange dashed rectangle in the top panel. Scale bars: 200 nm (top and middle) and 100 nm (bottom). (F) Selected frames from a live-cell recording of an oocyte injected with DiIC$_{18(3)}$ undergoing NE rupture. Scale bar: 1 µm. (G) STED image of the NE at the shell stage stained for NPCs (mAb414) (red), lamina (cyan) and phalloidin-AlexaFluor488 (gray). Scale bar: 1 µm.

microtubule-mediated tearing of the lamina (*Beaudouin et al., 2002*; *Salina et al., 2002*), but the precise morphology of the NE intermediates at a resolution comparable to that which we achieved in starfish oocytes is not known. Therefore, it is possible that a similar mechanism, based on

the segregation of lamina, NPCs and membranes, plays a critical role in destabilizing the NE during NE rupture in somatic cells.

One possibility is that membranes actually break (fission), but this is complicated by the fact that this process would probably require a coordinated fission of inner nuclear membranes. Another alternative mechanism of NE rupture may be the disassembly of the central ring of a nuclear pore. This could allow the nuclear envelope to rearrange into an ER-like network without the need for an actual change in membrane topology. It is clear that only a small subset of NPCs may undergo such complete disassembly, because we can visualize the majority of NPCs with still intact central ring structure long after rupture. F-actin assembly may facilitate this process by increasing membrane tension in evaginations, or by causing extreme membrane curvatures, for example at the boundary of invaginations, thereby facilitating NE rupture or possibly complete NPC disassembly.

In addition, although this hypothesis remains rather speculative at this point, it is intriguing to consider links between our work and what has been shown for repair of interphase NE rupture, as well as reassembly of NE after division, both instances involving the ESCRT machinery (*Denais et al., 2016*; *Raab et al., 2016*; *Olmos et al., 2015*). In our NEBD intermediates, we observe membrane topologies that are similar to those involved in NE repair, suggesting that the ESCRT complex may also be involved in these membrane rearrangements.

As mentioned in the 'Introduction', there is wide diversity across species and cell types in the extent of NE disassembly during division. For example, the lamina persists much longer in many species as compared to mammalian somatic cells (*Gruenbaum et al., 2003*). Intriguingly, an F-actin shell similar to that in starfish oocytes has been observed in several species, although these structures are yet to be characterized in detail. These species include other echinoderm species, such as sea urchin (*Burkel et al., 2007*), the cnidarian model *Nematostella vectensis* (*DuBuc et al., 2014*), and polychaete worms (*Jacobsohn, 1999*). These examples strongly suggest that F-actin-mediated NE rupture may be widely spread across animal species, and we speculate that the presence or absence of this mechanism may be correlated with differences in NE disassembly dynamics, nuclear size and other differences in physiology.

Finally, the clearing of NPCs off the membrane during NEBD appears to be conserved in organisms with partially open mitosis. Tearing of the NE has been observed in the fungus *Ustilago maydis* and the yeast *Schizosaccharomyces japonicus*. In *U. maydis*, the shearing is caused by microtubules pulling the nucleus through a small opening into the bud of the daughter cell (*Straube et al., 2005*). *Sz. japonicus,* on the other hand, splits the NE by stretching the nucleus between the two poles of a dividing cell (*Aoki et al., 2011*). Thus, NE rupture occurs by different means, but intriguingly, both show evidence of clearing NPCs before the NE is torn: *Sz. japonicus* redistributes the NPCs to the two poles, freeing naked membranes at the NE regions destined to be broken. NPCs of *U. maydis* on the other hand initiate release of nucleoporins prior to rupture just as in higher eukaryotes (*Straube et al., 2005*; *Aoki et al., 2011*; *Theisen et al., 2008*).

Taken together, our data suggest that the F-actin shell destabilizes the NE by segregating pore-dense and pore-free membranes, providing the first mechanistic explanation for the sudden collapse of the NE structure during its breakdown. As discussed above, this mechanism is likely to function in many animal species. In other species, forces may be generated by means other than Arp2/3-mediated actin polymerization, but the segregation of nuclear membranes from the lamin network appears to be a general feature of nuclear rupture observed in dividing mammalian somatic cells, as well as during the interphase NE rupture that is frequent in cancer cells.

## Materials and methods

**Key resources table**

| Reagent type or resource | Designation | Source or reference | Identifiers | Additional information |
|---|---|---|---|---|
| Biological sample (*Patiria miniata*) | starfish oocytes | https://scbiomarine.com/ http://www.marinusscientific.com/ | | |

*Continued on next page*

*Continued*

| Reagent type or resource | Designation | Source or reference | Identifiers | Additional information |
|---|---|---|---|---|
| Biological sample (*Patiria pectinifera*) | starfish oocytes | Kazuyoshi Chiba, Ochanomizu University, Tokyo, Japan | | |
| Sequence-based reagent | mEGFP3-UtrCH, 3mCherry-UtrCH | doi: 10.1002/cm.20226 | | synthetic mRNA (Utrophin CH domain (human)) |
| Peptide, recombinant protein | Importin | doi: 10.1093/emboj/16.6.1153 | | (Importin-ß (45-462)) |
| Peptide, recombinant protein | UtrCH | doi: 10.1002/cm.20226 | | (Utrophin CH domain (human)) |
| Antibody, mouse (mAb414) | mAb414, mouse monoclonal | Sigma, BioLegend | Sigma #N8786, Biologend #902907 | 1:250 |
| Antibody, rabbit (lamin) | lamin, rabbit polyclonal | see 'Materials and methods' for details | | 1:250 |
| Commercial assay or kit | AmpliCap-Max T7 High Yield Message Maker | CellScript | C-ACM04037 | |
| Commercial assay or kit | Poly(A) tailing kit | CellScript | C-PAP5104H | |
| Commercial assay or kit | Alexa Fluor 647 maleimide | ThermoFisher | A20347 | |
| Commercial assay or kit | Alexa Fluor 488 maleimide | ThermoFisher | A10254 | |
| Commercial assay or kit | Alexa Fluor 647 NHS Ester | ThermoFisher | A20006 | |
| Commercial assay or kit | Alexa Fluor 488 NHS Ester | ThermoFisher | A20000 | |
| Commercial assay or kit | Cy5 NHS Ester | discontinued | | |
| Chemical compound, drug | $DiIC_{18(3)}$ | ThermoFisher | D282 | |
| Chemical compound, drug | 1-methyladenine (1-MA) | ACROS organics | | |
| Commercial assay or kit | Phalloidin-Alexa Fluor 488 | ThermoFisher | A12379 | |
| Commercial assay or kit | Phalloidin-Alexa Fluor 568 | ThermoFisher | A12380 | |
| Commercial assay or kit | Phalloidin-Alexa Fluor 647 | ThermoFisher | A22287 | |
| Commercial assay or kit | Abberior STAR RED phalloidin | Abberior | | |
| Chemical compound, drug | CK-666 | Merck | 182515 | |
| Chemical compound, drug | Amino-dextran 500,000 MW | ThermoFisher | D7144 | |
| Chemical compound, drug | Amino-dextran 70,000 MW | ThermoFisher | D1862 | |
| Chemical compound, drug | TRITC–Dextran 155,000 MW | Sigma | T1287 | |

## Oocyte collection and injection

Starfish (*Patiria miniata* or *P. pectinifera*) were obtained in the springtime from Southern California (South Coast Bio-Marine LLC, Monterey Abalone Company or Marinus Scientific Inc) or were kindly provided by Kazoyushi Chiba (Ochanomizu University, Tokyo, Japan). They were kept at 16℃ for the

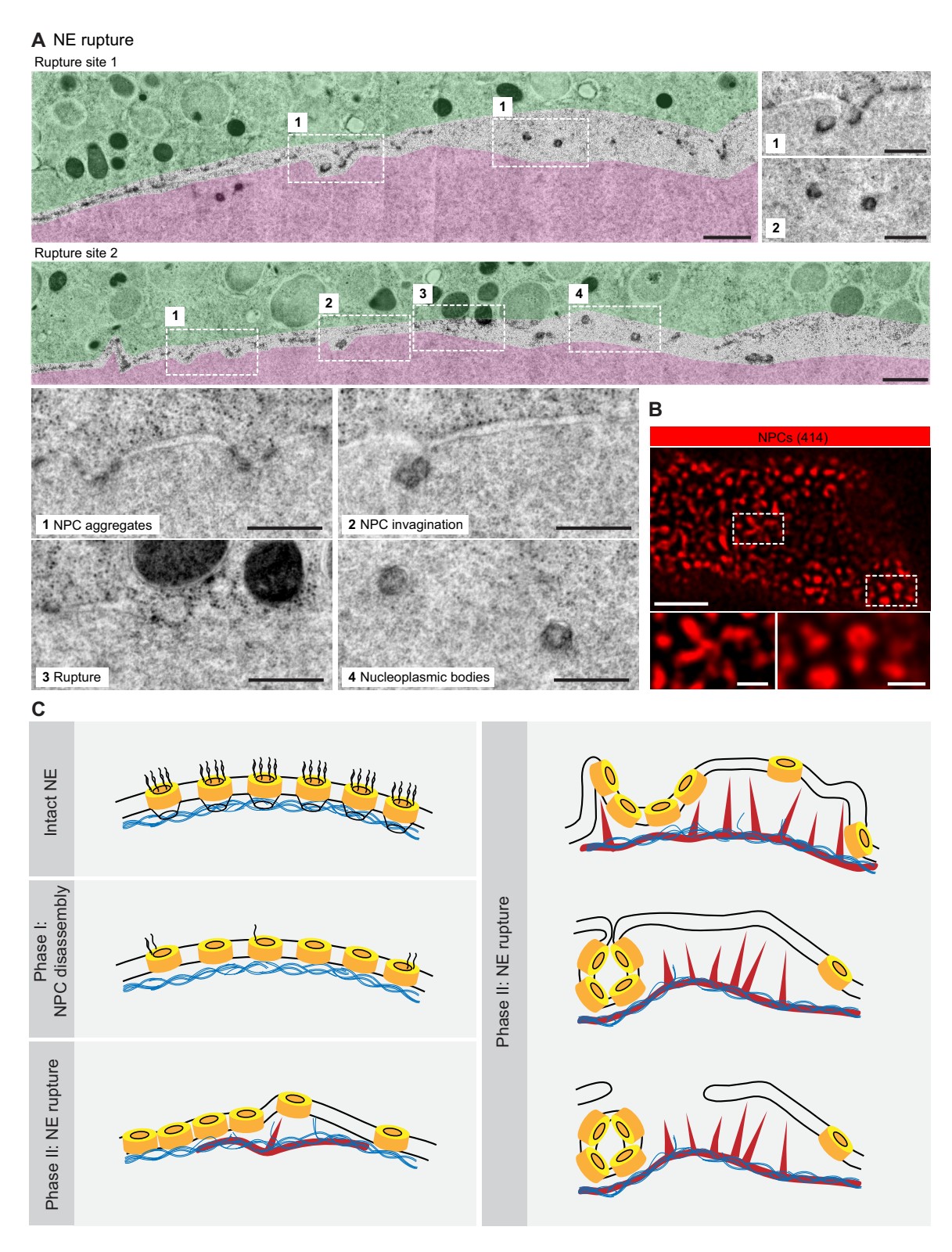

**Figure 7.** NE rupture occurs in pore-free regions. (**A**) Transmission EM images like those in **Figure 6A** showing examples of NE rupture events. Zooms of the areas outlined with white squares are shown below. Scale bars: 1 µm, and 500 nm for zooms. (**B**) *En face* STED image of the NE at the shell stage stained for NPCs (mAb414) (red). Scale bars: 1 µm (top) and 0.5 µm (bottom). (**C**) Model of F-actin-driven NE rupture. Intact NE: interphase organization of nuclear membranes (black lines) with regularly spaced NPCs (yellow cylinders) featuring cytoplasmic filaments and nuclear baskets. Nuclear baskets

*Figure 7 continued on next page*

*Figure 7 continued*

are embedded in the lamin network (blue filaments). Phase I of NEBD: peripheral NPC components are gradually released, but the NPC core and the overall NE structure remains intact. Phase II: NE rupture. First, small patches of F-actin (red) form within the lamina. F-actin patches grow and merge to form a shell, pushing apart NPCs that are still partially anchored in the lamina. As frequent F-actin spikes further sever the lamin-to-NE attachments, NPCs segregate into conglomerates, leaving stretches of unstable bare membrane, where breaks appear.

rest of the year in seawater aquariums at EMBL's or MPI-BPC's marine facilities. Oocytes were extracted from the animals fresh for each experiment as described earlier (*Lénárt et al., 2003*). mRNAs and other fluorescent markers were injected using microneedles, as described previously (*Jaffe and Terasaki, 2004*; *Borrego-Pinto et al., 2016*). mRNA was injected the day before to allow protein expression, whereas fluorescently labeled protein markers or dextrans were injected a few hours prior to imaging. Meiosis was induced at the initiation of the experiment by addition of 1-methyladenine (1-MA, 10 μM, Acros Organics). NEBD normally started 20–25 min after 1-MA addition, and only oocytes that initiated NEBD within 40 min of 1-MA addition were considered. Every experiment was repeated at least three times, with oocytes taken from at least two different animals.

## Fluorescent markers and antibodies

To label F-actin, 3mEGFP-UtrCH (*Burkel et al., 2007*) mRNA was synthesized in vitro from linearized DNA templates using the AmpliCap-Max T7 High Yield Message Maker kit (Cellscript), followed by polyA-tail elongation (A-Plus Poly(A) Polymerase Tailing Kit, Cellscript). mRNAs were dissolved in water (typical concentration 3–5 μg/μl) and injected into the oocyte up to 5% of the oocyte volume.

Alternatively, the UtrCH domain was cloned and expressed in *E. coli*, purified and labeled with Alexa Fluor 488- or 647-maleimide. Importin-β (45-462)-AlexaFluor488/647 protein was a generous gift from Dirk Görlich.

Phalloidin labeled with the indicated Alexa or Abberior fluorophores (Invitrogen) was dissolved in methanol, and was then air-dried prior to use and dissolved in PBS for immunostaining.

For dextrans, amino-dextrans were labeled with succinimidyl ester dye derivatives (Cy5 or Alexa-Fluor647) or purchased in already labeled form (TRITC), purified and injected into oocytes as described earlier (*Lénárt et al., 2003*). DiIC$_{18(3)}$ (ThermoFisher) was dissolved in sunflower oil to saturation and injected to oocytes.

For CK-666 (Merck) treatments, oocytes were treated at 0.5 mM final concentration and incubated for 1 hr prior to hormone addition.

The pan-NPC antibody mAb414 was purchased from BioLegend or Sigma (catalogue #902907 or #N8786, respectively). To produce the anti-starfish-lamin antibody, the *Patiria miniata* lamin sequence was first identified by BLAST searches in our transcriptome-based database by comparisons to the human lamin B amino acid sequence, and then confirmed by reverse searches to other species. Furthermore, the corresponding mRNA was expressed as a mEGFP fusion and showed the expected localization to the NE in starfish oocytes (not shown). Peptide antibodies were then produced against the 'histone-interaction peptide' region of starfish lamin (GTKRRRLDEEE SMVQSS), which was used as the antigen for rabbit immunization. Antibody production and affinity purification was performed by Cambridge Research Biochemicals. The antibody's specificity was confirmed by Western blots, which showed an expected-sized band, and by immunostaining, which showed localization to the nuclear rim in starfish oocytes.

## Immunostaining

Oocytes were fixed at the desired times in a PFA/GA fixative (100 mM HEPES [pH 7.0], 50 mM EGTA, 10 mM MgSO$_4$, 0.5% Triton-X100, 1 or 2% formaldehyde, 0.2 or 0.4% glutaraldehyde) modified from *Strickland et al. (2004)*. Active aldehyde groups that remained post fixation were quenched by 0.1% solution of NaBH$_4$ or by 200 mM NH$_4$Cl and 200 mM glycine. Subsequently, samples were permeabilized and blocked in PBS+0.1% Triton-X100 plus 3% BSA and the Image-IT reagent (ThermoFisher Scientific). Antibody staining was done overnight for the primary antibody and for 2–3 hr for the secondary antibody in PBS+0.1% Triton-X100 at room temperature. Oocytes were mounted with the antifade agent ProLongGold (ThermoFisher Scientific) under a coverslip pressed quite firmly onto tiny pillars of grease or double-sided tape (Scotch).

## Light microscopy

Live-cell movies were acquired on a Leica SP5 confocal microscope using a 40x HCX PL AP 1.10 NA water immersion objective lens (Leica Microsystems), or a Zeiss LSM880 AiryScan microscope using a C-Apochromat 40 × 1.20 NA water immersion objective lens. For live-cell imaging experiments, at least 3–5 oocytes were recorded per session.

Fixed oocytes were imaged on a Leica SP8 microscope equipped with the HC PL APO 1.40 NA 100x oil immersion objective according to Nyquist criteria. For STED imaging, suitable Abberior STAR 580 and Abberior STAR RED or Abberior STAR 635P secondary antibodies or nanobodies were used (Abberior, NanoTag). Samples were imaged on a Leica SP8 STED microscope, with the HC PL APO CS2 1.40 NA 100x oil immersion objective and using the 775 nm depletion laser. Alternatively, we used an Abberior Instruments STEDYCON scan head mounted onto a Nikon Ti2 microscope equipped with a 100x CFI Plan Apochromat Lambda NA 1.45 oil immersion objective lens, or with an Abberior Instruments Expert Line STED microscope using an Olympus 100x UPLSAPO 100XS NA 1.4 oil immersion objective. At least five oocytes were recorded per sample.

Live and fixed oocyte images were processed and deconvolved using the Huygens software (Scientific Volume Imaging) with either confocal, AiryScan or STED settings as appropriate.

## Electron microscopy

The electron microscopy protocol is described in detail in *Burdyniuk et al. (2018)*. In brief, oocytes were injected with a mixture of dextrans and a small batch was tested for meiosis timing. At the approximate time of NEBD, they were transferred into a carrier (three oocytes in 0.3 μl of sea water) and most of the water was removed with filter paper. Oocytes were immediately covered with a drop of 1-hexedecene, and immediately high-pressure frozen. Oocytes were freeze-substituted into Lowicryl HM-20. To stage the oocytes, light microscopy of EM sections was used to determine the progress of dextran entry. Selected sections were then post-stained with lead citrate and imaged using a BioTwin CM120 Philips transmission electron microscope at 120 kV. Large TEM montages were acquired using a JEOL JEM-2100Plus transmission electron microscope at 120 kV. Tomograms were reconstructed from tilt series acquired on a FEI Tecnai F30 transmission electron microscope at 300 kV with 1.554 nm pixel size.

More than 50 oocytes were processed by the above described method, and a few of these oocytes were found to be at the correct stage, showing NE rupture intermediates. The data shown in the figures are from a single oocyte that had perfect preservation throughout and has been sectioned through the entire nuclear volume. Of these sections, more than ten have been imaged and have been scanned as a large montage, and analyzed carefully. Three of these montages are shown as figure supplements.

## Acknowledgements

We thank the members of the Lénárt laboratory at EMBL for reagents and support, in particular Kálmán Somogyi, Andrea Callegari, Johanna Bischof, Joana Borrego-Pinto and Philippe Bun. We also thank EMBL's Advanced Light Microscopy Facility for essential support, specifically Marko Lampe for help with STED imaging. We thank the Electron Microscopy Core Facility, Yannick Schwab and Paolo Ronchi for sharing expertise during development of the EM protocol. We thank EMBL's Laboratory Animal Resources and Kresimir Crnokic in particular. We thank the members of the Lénárt group at MPI-BPC, in particular Jasmin Jakobi and Antonio Politi, as well as the staff of MPI-BPC's animal facility, in particular Ulrike Teichmann and Sascha Krause. We would like to thank Dirk Görlich and members of his laboratory for providing reagents and advice.

Research in PL's laboratory was funded by the European Molecular Biology Laboratory (EMBL) and the Deutsche Forschungsgemeinschaft (DFG) through grant GZ LE 2926/1–1 AOBJ 603520 of the Priority Programme SPP 1464. The laboratory is currently funded by the Max Planck Society.

## Additional information

### Funding

| Funder | Grant reference number | Author |
|---|---|---|
| Deutsche Forschungsgemeinschaft | SPP 1464 | Natalia Wesolowska |
| European Molecular Biology Laboratory | | Natalia Wesolowska<br>Pedro Machado<br>Celina Geiss<br>Hiroshi Kondo<br>Masashi Mori<br>Peter Lenart |
| Max Planck Society | | Ivan Avilov<br>Peter Lenart |

The funders had no role in study design, data collection and interpretation, or the decision to submit the work for publication.

### Author contributions

Natalia Wesolowska, Conceptualization, Data curation, Investigation, Visualization, Methodology, Writing - original draft, Writing - review and editing; Ivan Avilov, Investigation, Visualization, Writing - review and editing; Pedro Machado, Data curation, Investigation, Methodology; Celina Geiss, Data curation; Hiroshi Kondo, Masashi Mori, Investigation; Peter Lenart, Conceptualization, Data curation, Supervision, Funding acquisition, Investigation, Visualization, Methodology, Writing - original draft, Project administration, Writing - review and editing

### Author ORCIDs

Peter Lenart (iD) https://orcid.org/0000-0002-3927-248X

### Decision letter and Author response

Decision letter https://doi.org/10.7554/eLife.49774.sa1
Author response https://doi.org/10.7554/eLife.49774.sa2

## Additional files

### Supplementary files

• Transparent reporting form

### Data availability

Full resolution EM montages are provided as supplementary files.

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
