## [Decision Letter]

**Acceptance summary:**

The revised version of the manuscript is greatly improved. The most important improvement is the inclusion of Airy-scan/deconvolution experiments that provide simultaneous imaging of multiple key elements of nuclear envelope breakdown, including: (i) the nuclear envelope; (ii) filamentous actin; (iii) nuclear pore complexes; and (iv) nuclear permeability. Also the recasting of actin's role as 'prying the nuclear envelope away from the lamina' ties the data together into a more coherent story. The localization of the Arp2/3 complex to the base of spikes is also important, and the only (slight) disappointment is that no experiments were performed to try and identify the upstream nucleation promoting factor that turns on the Arp2/3 complex at the nuclear periphery. Overall this does a nice job of advancing our understanding of actin's role in driving nuclear envelope breakdown in starfish oocytes.

**Decision letter after peer review:**

Thank you for submitting your article "Nuclear envelope rupture in starfish oocytes proceeds by actin-driven segregation of pore-dense and pore-free membranes" for consideration by *eLife*. Your article has been reviewed by three peer reviewers, including R Dyche Mullins as the Reviewing Editor and Reviewer #1, and the evaluation has been overseen by Anna Akhmanova as the Senior Editor. The following individual involved in review of your submission has agreed to reveal their identity: William M. Bement (Reviewer #3).

The reviewers have discussed the reviews with one another and the Reviewing Editor has drafted this decision to help you prepare a revised submission.

Summary:

This manuscript presents novel interesting features of nuclear envelope breakdown at the onset of meiosis in starfish oocytes. The authors characterize the spatial relationships between the actin filaments, the nuclear envelope, the nuclear lamins, and the nuclear pore complexes and propose that actin assembly is needed for prying the nuclear envelope away from the underlying nuclear lamins. This mechanism represents a refinement of the previous view of actin's role in NEB in this system.

Essential revisions:

1) In several figures, such as Figure 4C, there are gaps in the organization of NPCs without deformation of the inner nuclear membrane (neither bumps nor spikes). The authors mention as well that "gaps appeared to evolve into bumps and membrane spikes". Doesn't this suggest that NPCs-free membrane areas are generated prior to actin spikes rather than "by F-actin-driven sorting of NE membranes into pore-dense and pore-free, ER-like membrane networks"? Is it possible that rupture occurs -as in other systems- via NPC disassembly, but that in these cells NPC disassembly is promoted by actin-dependent changes in membrane tension?

To strengthen the claim that actin assembly generates NPC-free membrane protrusions the authors should provide EM images of the envelope under conditions where the shell is prevented from forming. The fluorescence images are not sufficient, since the membrane cannot be seen. In this context, it would be useful to see the NE independently of NPCs, e.g. using ER tracker or WGA.

2) The authors conclude that the pore-free membrane is unstable and thus serves as the site of ruptures. There is, however, no evidence that the ruptures arise first at the points of maximum membrane delamination. The authors need to address this point by determining whether the rupture occurs at the tips of spikes, at the shoulders of spikes, or elsewhere.

3) The authors need to marshal quantitative support for their main conclusions. It appears that there is just a single image in the paper that shows actin filaments inside one of the membrane delaminations (Figure 4E). Thus, it is important to know just how frequent this phenomenon is. This point is particularly important given the fact that the authors have shown that when F-actin assembly is suppressed by CK666, the nuclear envelope nonetheless becomes uneven. Other points that would benefit from quantification include the relative density of nuclear pores on the shoulders of delamination sites.

For your information, we include the complete reviews below, but please focus on the revision points listed above.

Reviewer #1:

This manuscript presents previously undescribed features of nuclear envelope breakdown at the onset of meiosis in starfish oocytes. Using these same cells the authors previously uncovered a role for actin filaments in confining chromosomes and facilitating their capture by spindle microtubules. They also demonstrated that nuclear envelope breakdown in starfish oocytes requires transient assembly of a branched actin network on the inner surface of the nuclear envelope (Mori et al., 2014). The 2014 Current Biology paper reported the presence of radially oriented actin bundles, or spikes, growing out of a lamin-associated actin network and into the nuclear envelope. At the time the authors interpreted the role of these spikes as puncturing or forcefully fragmenting the nuclear envelope.

In the current work the authors have employed higher resolution light microscopy techniques to characterize the spatial relationships between the actin filaments, the nuclear envelope, the nuclear lamins, and the nuclear pore complexes. The major result is that the role of actin assembly appears to be 'delamination' of the nucleus by prying the nuclear envelope away from the underlying nuclear lamins. This is not a brand new mechanism but certainly represents a refinement of the previous view of actin's role in NEB in this system. I am inclined to think it justifies publication as a brief report, but there are a few points that should be addressed before publication.

1) The authors previously localized the Arp2/3 complex to the nuclear actin shell (in the 2014 Curr Biol paper), but they show no Arp2/3 localization in the current paper. They should use their super-resolution light microscopy to investigate the relationship between Arp2/3, filamentous actin, and the nuclear envelope. My suspicion is that the Arp2/3 will be associated with the lamins, but not with the actin spikes or the overlying nuclear envelope.

2) Which nucleation promoting factors are activating the Arp2/3 complex in this system? The obvious candidate is WASP. Do any WASP antibodies cross-react with the starfish protein?

3) The filopodia-like morphology of the actin spikes suggests that perhaps some nuclear envelope-associated polymerase is promoting filament elongation. Have the authors looked for VASP et al. on the inner surface of the nuclear envelope?

Reviewer #2:

In this manuscript, Wesolowska et al. use super-resolution light and electron microscopy in combination to study how actin contributes to nuclear envelope rupture in starfish oocytes. This builds upon previous work from the team published in Current Biology that described an important role for Arp2.3 nucleated actin spikes in NEB.

In their new paper, the authors study the ultrastructure in more detail. They show that nuclear envelope is uniform and densely packed with NPCs in immature oocytes, before becoming heterogeneous with the generation of actin spikes, when it breaks up into regions that have a high or low density of NPCs prior to nuclear envelope rupture. The authors argue that the actin filaments nucleated by the Arp2/3 complex embedded within the nuclear lamina push on the nuclear envelope to induce membrane deformations (bumps and spikes), which lead to the separation of NPCs-free and NPCs-rich membranes and to subsequent ruptures of the envelope.

The paper is well written, and the data are presented clearly. There is some valuable new information about how the ultrastructure of the nuclear envelope evolves from immature oocyte to the NEBD with a complete rupture of the nuclear envelope. Some questions remain, however, about precisely how actin spikes contributes to rupture.

In the model depicted in Figure 6, the key events are 1) A mitotic specific change in NPCs 2) The actin-dependent partitioning of NPCs into rich and poor regions. 3) Rupture.

This series of events is not exactly demonstrated in the data:

1) In several figures, such as Figure 4C, there are gaps in the organization of NPCs without deformation of the inner nuclear membrane (bumps neither spikes). The authors mention as well that "gaps appeared to evolve into bumps and membrane spikes". Doesn't this suggest that NPCs-free membrane areas are generated prior to actin spikes rather than "by F-actin-driven sorting of NE membranes into pore-dense and pore-free, ER-like membrane networks".

2) The paper doesn't do enough to explain how changes in NPCs themselves contribute to this process. Where does NPC re-modelling come into the picture?

3) The link between membrane rupture and local actin spikes remains a bit unclear. In images, the most obvious effect of an actin spike is the separation of inner and outer envelopes. One might then expect rupture within spike regions to sometimes break one membrane – which would seem a very bad thing to do. Isn't it more likely that in the end rupture occurs via NPC disassembly as in other systems – but that this is membrane tension (and therefore actin) dependent ?

Reviewer #3:

Nuclear envelope breakdown is a fundamental feature of the cell cycle in most animal cells and yet its basis remains poorly understood. In the textbook view, it somehow arises as a consequence of phosphorylation of lamins and components of the nuclear envelope such as the nucleoporins. There is also evidence that cell cycle-regulated, microtubule motor-dependent transport can tear the envelope in some cell types and that polymerizing actin can tear it in other cell types. But in neither case, has the process been studied in anywhere near sufficient detail for anything more than vague conclusions to be drawn.

In the current study, using a careful combination of light and electron microscopy, the authors have obtained evidence that during the disassembly of the germinal vesicle (the large oocyte nucleus) in starfish, spikes of actin that assemble from within the nuclear lamina drive evagination of the nuclear envelope concomitant with displacement of the nuclear pore complexes. That is, as the spikes extend outward (i.e. toward the cytoplasm) the nuclear pore complexes are left behind in patches flanking the spikes while the nuclear envelope overlying the spikes is nuclear pore complex-free. The authors propose that the evaginations of pore-free envelope are unstable and consequently rupture.

This is fascinating, and beautifully executed work. However, the authors should address the following points before their conclusions can be accepted with confidence.

1) The authors need to marshal quantitative support for their main conclusions. As far as I can tell, there is just a single image in the paper that shows actin filaments inside one of the evaginations (Figure 4E). Thus, it is important to know just how frequent this phenomenon is. This point is particularly important given the fact that the authors have shown that when F-actin assembly is suppressed by CK666, the nuclear envelope nonetheless becomes uneven. Other points that would benefit from quantification include the relative density of nuclear pores in evaginations versus the shoulders of evaginations.

2) The authors state "Our EM data clearly show that the F-actin shell protrudes pore-free nuclear membranes, separating these from the lamina". This is not quite true as there are no EM data showing the envelope under conditions where the shell is prevented from forming. Such data should be provided. The fluorescence imaging does not allow this point to be tested as the membrane cannot be seen.

3) The authors also conclude that the pore-free membrane is unstable and thus serves as the site of ruptures. However, unless I missed it, there is no evidence that the ruptures arise first at the points of the evaginations. Given that one of the goals of the study is to understand how rupturing occurs, the authors need to address this point by determining whether the rupture occurs at the tips of spikes, at the shoulders of spikes, or elsewhere.

---

## [Author Response]

Essential revisions:1) In several figures, such as Figure 4C, there are gaps in the organization of NPCs without deformation of the inner nuclear membrane (neither bumps nor spikes). The authors mention as well that "gaps appeared to evolve into bumps and membrane spikes". Doesn't this suggest that NPCs-free membrane areas are generated prior to actin spikes rather than "by F-actin-driven sorting of NE membranes into pore-dense and pore-free, ER-like membrane networks"? Is it possible that rupture occurs -as in other systems- via NPC disassembly, but that in these cells NPC disassembly is promoted by actin-dependent changes in membrane tension?To strengthen the claim that actin assembly generates NPC-free membrane protrusions the authors should provide EM images of the envelope under conditions where the shell is prevented from forming. The fluorescence images are not sufficient, since the membrane cannot be seen. In this context, it would be useful to see the NE independently of NPCs, e.g. using ER tracker or WGA.

To begin at the end, while we agree that visualizing the NE by EM in the absence of the F-actin shell would be a critical experiment, unfortunately, this is technically not feasible. The reason is that our correlative approach uses the F-actin shell or the rapid entry of a large dextran as temporal reference. If the F-actin shell is inhibited, we lose both of these references, and thus we are unable to time our samples, which is essential to capture *intermediates* of this fast process. In such oocytes we are only able to visualize the end-point, which we already did in our previous study (Mori et al., 2014). There we showed both by electron and light microscopy that in the absence of the F-actin shell large pieces of nuclear envelope remain intact up 10 minutes after NEBD.

As an alternative, we imaged NE intermediates in live oocytes. To our delight, by combining Zeiss’ AiryScan technology with deconvolution we were able to resolve membrane reorganization during NE rupture in live oocytes (which we previously thought is not feasible). In addition to DiIC_18(3),_which provided bright membrane labeling, we used a set of recombinant-protein-based probes, namely UtrCH-Alexa488/647 to label F-actin, a fragment of Importin-β coupled to Alexa488/647 to label nuclear pores, as well as a large, 500 kDa dextran as a permeability probe.

First, with this assay in hand, we were able to show that if the F-actin shell is prevented from forming, membrane reorganization at the fine, micrometer scale does not occur, and large sections of the membrane remain intact (new Figure 1E), consistent with our previous study (Mori et al., 2014). These data also clarify the important point that while in the absence of the F-actin shell membrane reorganization at the fine scale (i.e. NE rupture) does not occur, on the larger scale the nucleus does collapse and the NE folds due to the gradual mixing of cyto- and nucleoplasm. (Note that in these live-cell experiments we used high amounts of UtrCH to prevent F-actin shell assembly. As we showed previously (Mori et al., 2014), depleting available monomers is an effective means to prevent Arp2/3-mediated actin assembly. We have taken this approach, because while we obtained essentially the same results using the Arp2/3 inhibitor CK-666, these experiments were not fully reproducible. We think that the reason for this is that CK-666 is sensitive to light (it is green), especially at the high laser powers required for these high resolution recordings.)

Second, in live cells we were able to directly record and thus firmly establish the temporal order of events during NE rupture (new Figure 1B-D). The process lasts about 1 minute and is initiated by actin assembly on the inner side of the NE. At this point no NE rearrangements can be seen either by DiIC_18(3)_ or the NPC marker, Importin-β. Membrane rearrangements are only first detected approx. 15-20 s after actin assembly has started. Rupture occurs last, at approx. 30-40 s after the start of actin assembly. Thus, our live cell data clearly indicate that actin assembly starts first, and is followed by subsequent reorganization of the NE.

Third, we actually agree with the reviewers, and we now mention this in the Discussion that rupture may be triggered by completedisassembly of a subset of NPCs. However, our data also show that this complete NPC disassembly may only affect a small subset of NPCs, as we can see many NPCs with intact core-ring structure long after rupture. As we show, rupture is facilitated by the F-actin shell. Here our primary hypothesis is that the F-actin shell facilitates rupture by segregating pore-free and pore-dense regions. It is likely that this will have an effect on membrane tension, which may indeed contribute to facilitate rupture. We thank reviewer 2 for the proposal, which we now mention in the Discussion.

2) The authors conclude that the pore-free membrane is unstable and thus serves as the site of ruptures. There is, however, no evidence that the ruptures arise first at the points of maximum membrane delamination. The authors need to address this point by determining whether the rupture occurs at the tips of spikes, at the shoulders of spikes, or elsewhere.

To better address this point, we acquired montages of several additional EM sections and carefully examined again all of our previous and these new EM data. We consistently observed at least 5 rupture sites that: *(i)* in ruptured regions, in which no continuous membrane boundary between cytoplasm and nucleus is present, nucleoplasmic bodies / NPC conglomerates are abundant; *(ii)* directly neighboring ruptured regions the NE is segregated into pore-free and pore-dense segments. While pore-dense regions show various stages of invagination into the nucleoplasm, pore-free regions evaginate. Thus, the main difference between ruptured and directly neighboring still-intact regions is that in intact regions pore-free segments connect pore-dense segments, while in ruptured regions these segments connecting pore-dense regions are absent. These observations confirm our previous conclusion that rupture occurs in pore-free segments evaginating between pore-dense segments.

Admittedly, these steps were not sufficiently illustrated in the original manuscript. We now show a different crop of the previously shown ruptured region also including the neighboring intact area (now Figure 7A, rupture site 1). Additionally, we show another ruptured region, which we identified on a different section distal from the other site in the same oocyte (Figure 7A, rupture site 2). In our opinion, these two sites illustrate well the above features typical to rupture sites. Together, we hope that these new panels illustrate the event of rupture in a much better and much more convincing way as compared to our original submission.

3) The authors need to marshal quantitative support for their main conclusions. It appears that there is just a single image in the paper that shows actin filaments inside one of the membrane delaminations (Figure 4E). Thus, it is important to know just how frequent this phenomenon is. This point is particularly important given the fact that the authors have shown that when F-actin assembly is suppressed by CK666, the nuclear envelope nonetheless becomes uneven. Other points that would benefit from quantification include the relative density of nuclear pores on the shoulders of delamination sites.

The original manuscript already contained a panel (now Figure 4C) indicating the categories of observed rupture intermediates, and the number of times every intermediate was observed per section across the nucleus (for the section shown and in brackets showing numbers counted on two additional sections shown in Figure 4—figure supplement 2 and 3, respectively). The original manuscript also already contained a quantification of pore densities, which was shown in a supplementary figure and now moved to the main Figure 6C.

To additionally address these points, we acquired montages of several more EM sections, and, as mentioned above, we recorded additional live-cell data. Generally, combining these two types of data strengthened our conclusions in many ways relevant to this and other points. First, the live cell data directly visualize and thus firmly establish the temporal order of events. Second, while EM is by nature a technique of low sample numbers, we were able to record live videos of many oocytes, which gives us much more confidence and strong support to our conclusions. To the specific points:

i) We included additional examples of actin filaments visible in membrane evaginations seen on electron micrographs (Figure 5—figure supplement 1A). We see no easy way to further quantify these observations because actin filaments are generally difficult to see by EM. They are only visible if they lay parallel to the plane of sectioning, the section has to lay perfectly on the grid, and the sample has to be perfectly focused (which is an issue when acquiring montages automatically). Furthermore, there are also slight local differences in the quality of preservation, which is unfortunately typical to high-pressure frozen large samples.

ii) We show in live oocytes that membrane evaginations form abundantly in untreated oocytes, while the membrane remain smooth when the F-actin shell is inhibited (new Figure 1D, E). These data also clarify the slightly confusing fact that, while in the absence of the F-actin shell the NE does not rupture on the fine, micrometer scale, even in the absence of the F-actin shell the NE collapses and folds at the larger spatial scale (tens of microns). Importantly, we were also able to directly visualize F-actin in membrane evaginations in live oocytes (new Figure 5F).

iii) We show additional electron micrographs of NPCs in intact regions and near rupture sites (Figure 6D), and we moved the quantification of NPC densities previously shown in the supplement to Figure 6C.

For your information, we include the complete reviews below, but please focus on the revision points listed above.Reviewer #1:[…] There are a few points that should be addressed before publication.1) The authors previously localized the Arp2/3 complex to the nuclear actin shell (in the 2014 Curr Biol paper), but they show no Arp2/3 localization in the current paper. They should use their super-resolution light microscopy to investigate the relationship between Arp2/3, filamentous actin, and the nuclear envelope. My suspicion is that the Arp2/3 will be associated with the lamins, but not with the actin spikes or the overlying nuclear envelope.

Yes, indeed. We performed immunostainings with our previously developed starfish ArpC1 antibody and found that Arp2/3 is localized at the base of the F-actin shell, slightly below the layer of NPCs. The spikes are not labeled by ArpC1 (Figure 3E)

2) Which nucleation promoting factors are activating the Arp2/3 complex in this system? The obvious candidate is WASP. Do any WASP antibodies cross-react with the starfish protein?

Unfortunately, while we tested several commercially available antibodies, we did not find any, which would cross-react with starfish WASP.

3) The filopodia-like morphology of the actin spikes suggests that perhaps some nuclear envelope-associated polymerase is promoting filament elongation. Have the authors looked for VASP et al. on the inner surface of the nuclear envelope?

Unfortunately, the answer is again no. As it appears, we will need to identify WASP, VASP and other related regulators in the starfish genome/transcriptome, which we are planning to combine with proteomic analyses of oocytes and eggs in order to nail down potential candidates. We will then need to generate starfish-specific reagents. We think that these are major efforts beyond the scope of the present manuscript.

Nevertheless, we are very excited to follow up on the detailed molecular mechanism in the near future, because even if it tuned out that this mechanism is specific to facilitate NE rupture in echinoderm oocytes, it could be very interesting to try to “transplant” this mechanism to other cells. For example, to artificially induce NE rupture in somatic mammalian cells.

Reviewer #2:[…] The paper is well written, and the data are presented clearly. There is some valuable new information about how the ultrastructure of the nuclear envelope evolves from immature oocyte to the NEBD with a complete rupture of the nuclear envelope. Some questions remain, however, about precisely how actin spikes contributes to rupture.In the model depicted in Figure 6, the key events are 1) A mitotic specific change in NPCs 2) The actin-dependent partitioning of NPCs into rich and poor regions. 3) Rupture.This series of events is not exactly demonstrated in the data:1) In several figures, such as Figure 4C, there are gaps in the organization of NPCs without deformation of the inner nuclear membrane (bumps neither spikes). The authors mention as well that "gaps appeared to evolve into bumps and membrane spikes". Doesn't this suggest that NPCs-free membrane areas are generated prior to actin spikes rather than "by F-actin-driven sorting of NE membranes into pore-dense and pore-free, ER-like membrane networks".

To address these points, we recorded new live-cell data. We were very satisfied that by combining Zeiss’ AiryScan technology with deconvolution were able to resolve membrane reorganization during NE rupture in live oocytes. In addition to DiIC_18(3),_which provided bright membrane labeling, we used a set of recombinant protein based probes, namely UtrCH-Alexa488/647 to label F-actin, a fragment of Importin-β coupled to Alexa488/647 to label nuclear pores, as well as a large, 500 kDa dextran as a permeability probe.

With this assay in hand, we were able to directly record and thus firmly establish the temporal order of events during NE rupture (new Figure 1.). The process last about 1 minute in total and is initiated by actin assembly on the inner side of the NE. At this point no NE rearrangements are seen visualized either by DiIC_18(3)_ or Importin-β. Membrane rearrangements are first detected approx. 15-20 s after actin assembly has started. Rupture occurs last, at approx. 30-40 s. Thus our live cell data clearly suggest that actin assembly is first, followed by subsequent reorganization of the NE.

As an additional note, while in somatic cells the NE is a stably cross-linked structure with the NPCs stably embedded in the lamina (Daigle et al., 2001), this may not necessarily be the case in the starfish oocyte just before NE rupture. By this time, due to the preceding phosphorylation-driven partial disassembly of NPCs, several NPC and NE components have already been released (Lénárt et al., 2003). This, for example, includes Nup153, which has been shown in somatic cells to be required to anchor NPCs to the lamina (Daigle et al., 2001). Furthermore, it has been shown that in *Drosophila* embryos NPC-lamina cross-links are generally absent until later stages of development (Hampoelz et al., 2016). Together, these data suggest that NPCs may be mobile within the NE of the starfish oocyte at this stage, in which case they can be easily pushed aside by actin assembly.

2) The paper doesn't do enough to explain how changes in NPCs themselves contribute to this process. Where does NPC re-modelling come into the picture?

As we published earlier, NE rupture is preceded by partial NPC disassembly, which renders NPCs ‘leaky’, but leaves the core NPC ring and the overall NE structure intact (Lénárt et al., 2003). The very good preservation of ultrastructure in our high-pressure frozen EM samples confirms this point, as we can visualize NPCs with intact core ring structures in still intact NE regions neighboring rupture sites (new Figure 6D). That these pores are ‘leaky’, i.e. that several peripheral nucleoporins have been released at this stage, is unfortunately not directly visible at the ultrastructural level.

Functionally, we do think that partial NPC disassembly is critical and serves at the trigger for subsequent NE rupture. This is not the topic of the current study, but in our previous works (Mori et al., 2014; Lénárt et al., 2003) we have shown by using dextran fractions of different sizes as well as fluorescent nuclear import and export cargoes that NE rupture commences when partial NPC disassembly reaches a certain threshold. Although the specific molecules remain still to be identified, these findings suggest that a set of cytoplasmic proteins start entering the nucleus during partial NPC disassembly, and actin assembly and subsequent rupture is triggered when these proteins reach a critical nuclear concentration. When we injected actin or Arp2/3 into the nucleus of oocytes, this was not sufficient to trigger F-actin shell formation, which further suggest that the critical component may be an activator of Arp2/3 or another upstream component.

3) The link between membrane rupture and local actin spikes remains a bit unclear. In images, the most obvious effect of an actin spike is the separation of inner and outer envelopes. One might then expect rupture within spike regions to sometimes break one membrane – which would seem a very bad thing to do. Isn't it more likely that in the end rupture occurs via NPC disassembly as in other systems – but that this is membrane tension (and therefore actin) dependent ?

We agree with the reviewer and we amended the Discussion to clarify these points. First, it is very clear in our EM data that NPCs serve as the spacer to keep outer and inner nuclear membranes at a fixed distance. In NPC-free regions the spacing between outer and inner membranes becomes irregular. However, we did not observe a single instance of rupture of only one of the two membranes – which we agree that would seem not only a bad thing to do, but also rather unlikely to happen.

However, we now mention in the Discussion that rupture may be triggered by completedisassembly of NPCs. However, our data also clearly suggest that this complete NPC disassembly only affects a small subset of NPCs, as we can see many NPCs with intact core-ring structure even long after rupture. As we also show, rupture is facilitated by the F-actin shell. Here our primary hypothesis is that the F-actin shell facilitates rupture by segregating pore-free and pore-dense regions, as explained above. It is very likely that this will have an effect on membrane tension, which may indeed contribute to facilitating rupture. We thank the reviewer for the proposal and we now mention this hypothesis in the Discussion.

Reviewer #3:[…] This is fascinating, and beautifully executed work. However, the authors should address the following points before their conclusions can be accepted with confidence.1) The authors need to marshal quantitative support for their main conclusions. As far as I can tell, there is just a single image in the paper that shows actin filaments inside one of the evaginations (Figure 4E). Thus, it is important to know just how frequent this phenomenon is. This point is particularly important given the fact that the authors have shown that when F-actin assembly is suppressed by CK666, the nuclear envelope nonetheless becomes uneven. Other points that would benefit from quantification include the relative density of nuclear pores in evaginations versus the shoulders of evaginations.

To address these points, we acquired montages of several additional EM sections, and, as mentioned above, we recorded additional live-cell data. Generally, combining these two types of data strengthened our conclusions in many ways relevant to this and other points. First, the live cell data directly visualize and thus firmly establish the temporal order of events. Second, while EM is by nature a technique of low sample numbers, we were able to record live videos of many oocytes, which gives us much more confidence and strong support to our conclusions. To the specific points:

1) We included additional examples of actin filaments visible in membrane evaginations seen on electron micrographs (Figure 5—figure supplement 1A). We see no easy way to further quantify these observations because actin filaments are generally difficult to see by EM. They are only visible if they lay parallel to the plane of sectioning, the section has to lay perfectly on the grid, and the sample has to be perfectly focused (which is an issue when acquiring montages automatically). Furthermore, there are also slight local differences in the quality of preservation, which is unfortunately typical to high-pressure frozen large samples.

2) We show in live oocytes that membrane evaginations form abundantly in untreated oocytes, while the membrane remain smooth when the F-actin shell is inhibited (new Figure 1D, E). These data also clarify the slightly confusing fact that, while in the absence of the F-actin shell the NE does not rupture on the fine, micrometer scale, even in the absence of the F-actin shell the NE collapses and folds at the larger spatial scale (tens of microns). Importantly, we were also able to directly visualize F-actin in membrane evaginations in live oocytes (new Figure 5F).

3) We show additional electron micrographs of NPCs in intact regions and near rupture sites (Figure 6D), and we moved the quantification of NPC densities previously shown in the supplement Figure 6C.

2) The authors state "Our EM data clearly show that the F-actin shell protrudes pore-free nuclear membranes, separating these from the lamina". This is not quite true as there are no EM data showing the envelope under conditions where the shell is prevented from forming. Such data should be provided. The fluorescence imaging does not allow this point to be tested as the membrane cannot be seen.

While we agree that this would have been a critical experiment, we were unfortunately unable to capture the corresponding stage in shell-inhibited oocytes by electron microscopy. The reason is that our correlative approach uses the F-actin shell or the rapid entry of a large dextran as temporal reference. If the F-actin shell is inhibited, we lose both of these references, and thus we are unable to time our samples, which critical to visualize *intermediates* of this fast process. In such oocytes we are only able to visualize the end-point, which we already did in our previous study (Mori et al., 2014). There we showed both by electron and light microscopy that in the absence of the F-actin shell large pieces of nuclear envelope remain intact up 10 minutes after NEBD.

As mentioned above, as an alternative approach we established imaging of membrane intermediates in live oocytes. These data clearly show that: (i) when the F-actin shell is prevented from forming, the membrane remains smooth and no evaginations form (new Figure 1E); (ii) F-actin spikes are directly visualized in live cells to protrude membranes (new Figure 5F).

3) The authors also conclude that the pore-free membrane is unstable and thus serves as the site of ruptures. However, unless I missed it, there is no evidence that the ruptures arise first at the points of the evaginations. Given that one of the goals of the study is to understand how rupturing occurs, the authors need to address this point by determining whether the rupture occurs at the tips of spikes, at the shoulders of spikes, or elsewhere.

To better address this point, we acquired montages of several additional EM sections and carefully examined again all of our previous and these new EM data. We consistently observed at least 5 rupture sites that: (i) in ruptured regions, in which no continuous membrane boundary between cytoplasm and nucleus is present, nucleoplasmic bodies / NPC conglomerates are abundant; (ii) neighboring ruptured regions the NE is segregated into pore-free and pore-dense segments. While pore-dense regions show various stages of invagination into the nucleoplasm, pore-free regions evaginate. Thus, the difference between ruptured and directly neighboring still-intact regions is that in intact regions pore-free segments connect the pore-dense segments, while in ruptured regions these segments connecting pore-dense regions are absent. These observations confirm our previous conclusion that rupture occurs in pore-free segments evaginating between pore-dense segments.

Admittedly, these steps were not sufficiently illustrated in the original manuscript. We now show a different crop of the previously shown ruptured region also including the neighboring intact area (now Figure 7A, rupture site 1). Additionally, we show another ruptured region, which we identified on a different section and distal from the other site in the same oocyte (Figure 7A, rupture site 2). The two sites show all the above detailed features we found typical to rupture sites. Together, we hope that these new panels illustrate the event of rupture in a much better and much more convincing way as compared to our original submission.